# Recognition of galactose by a scaffold protein recruits a transcriptional activator for the *GAL* regulon induction in *Candida albicans*

Xun Sun[1], Jing Yu[2], Cheng Zhu[3], Xinreng Mo[2], Qiangqiang Sun[2], Dandan Yang[1], Chang Su[2]\*, Yang Lu[1]\*

[1]College of Life Sciences, TaiKang Center for Life and Medical Sciences, Wuhan University, Wuhan, China; [2]Hubei Key Laboratory of Cell Homeostasis, Wuhan University, Wuhan, China; [3]Tianjin Key Laboratory of Function and Application of Biological Macromolecular Structures, Tianjin University, Tianjin, China

\*For correspondence:
changsu@whu.edu.cn (CS);
ylu7@whu.edu.cn (YL)

**Competing interest:** The authors declare that no competing interests exist.

**Abstract** The *GAL* pathway of yeasts has long served as a model system for understanding of how regulatory mode of eukaryotic metabolic pathways evolves. While Gal4 mode has been well-characterized in Saccharomycetaceae clade, little is known about the regulation of the *GAL* pathway in other yeasts. Here, we find that Rep1, a Ndt80-like family transcription factor, serves as a galactose sensor in the commensal-pathogenic fungus *Candida albicans*. It is presented at the *GAL* gene promoters independent of the presence of galactose. Rep1 recognizes galactose via a direct physical interaction. The net result of this interaction is the recruitment of a transcriptional activator Cga1 (Candida galactose gene activator, orf19.4959) and transcription of the *GAL* genes proceeds. Rep1 and Cga1 are conserved across the CTG species. Rep1 itself does not possess transcriptional activity. Instead, it provides a scaffold to recruit different factors for transcriptional regulation. Rep1-Cga1 mode of regulation represents a new example of network rewiring in fungi, which provides insight into how *C. albicans* evolves transcriptional programs to colonize diverse host niches.

## Editor's evaluation

This important manuscript investigates the circuitry connecting the galactose utilization regulon of the human pathogen and model organism *Candida albicans* to the sensing of galactose. In the non-pathogenic model yeast *Saccharomyces cerevisiae* this circuit represents a textbook model that rivals the lac operon as a teaching tool. Using a broad array of mainly classical approaches, this study convincingly demonstrates the transcriptional activators that are required for galactose responsive metabolic genes in *C. albicans*. The recognition of just how different the regulation of the galactose pathway across fungal species represents an important advance in our understanding of the evolution of the regulatory control of these circuits, and makes a nice addition to the textbook version of eukaryotic gene regulation.

## Introduction

Galactose is a monosaccharide that is abundant in nature and is found in many forms. The *GAL* pathway of the budding yeast *Saccharomyces cerevisiae* is a favorite textbook example of the regulation of gene expression in eukaryotes. This pathway is controlled through the Gal4 activator, a

member of the Cys6 zinc finger class of transcription factors found only in fungi (*Burger et al., 1991*), as well as a sensor (Gal3) and a repressor (Gal80). In the absence of galactose, Gal80 prevents transcriptional activation of the pathway by means of a physical interaction with Gal4 (*Platt and Reece, 1998*). In the presence of galactose, Gal3 relieves this repression of Gal4 (*Sil et al., 1999*; *Peng and Hopper, 2000*; *Sellick and Reece, 2006*), thereby activating the transcription of target *GAL* genes such as *GAL1*, *GAL7*, and *GAL10* (*Johnston, 1987*). The enzymatic products of three Leloir genes act sequentially to convert galactose to glucose-6-phosphate, which enters glycolysis.

The *GAL* pathway of budding yeasts is a powerful model for inferring the evolutionary principles guiding the evolution of eukaryotic metabolic and genetic pathways (*Harrison et al., 2022*; *LaBella et al., 2021*; *Hittinger et al., 2010*; *Slot and Rokas, 2010*; *Hittinger et al., 2004*). Although the Gal3-Gal80-Gal4 mode in *S. cerevisiae* is a favorite textbook example of the regulation of gene expression in eukaryotes (*Johnston, 1987*; *Campbell et al., 2008*; *Conrad et al., 2014*; *Giniger et al., 1985*; *Traven et al., 2006*), we know unexpectedly little about the regulation of the *GAL* pathway in budding yeasts other than the Saccharomycetaceae clade. The DNA-binding domain encoded by *ScGAL4* is conserved throughout the budding yeast subphylum, but its Gal80-binding domain is only conserved in Saccharomycetaceae whose pattern of conservation mirrors that of *GAL80* (*Traven et al., 2006*; *Pan and Coleman, 1989*), arguing that the Gal4-Gal80 mode of regulation is restricted to the Saccharomycetaceae clade. *Candida albicans* exists primarily as a commensal resident of the GI tract of humans. However, it can infect sites ranging from the skin and the oral and vaginal mucosa to deep tissues from colonization of the patient's own GI tract if host or environmental factors are permissive (*Brown et al., 2012*). Although the *GAL* regulon structure and the overall logic of regulation have been preserved (*Brown et al., 2009*; *Fitzpatrick et al., 2010*), there are significant differences in the quantitative response of these genes to galactose between *S. cerevisiae* and *C. albicans*. For example, the *GAL* genes induction is faster and more sensitive to galactose concentration in *C. albicans* compared to that of *S. cerevisiae* (*Dalal et al., 2016*). These findings raise the question of how the gene regulatory network of *GAL* regulon has evolved in *C. albicans* to generate such biological diversity since orthologs of Gal80 and Gal3 are absent in *C. albicans*.

Considerable efforts have been made to identify numerous transcription factors involved in the galactose-inducible expression of three *GAL* genes in *C. albicans* (*Brown et al., 2009*; *Dalal et al., 2016*; *Martchenko et al., 2007*), none of which, however, are necessary for galactose utilization in this fungus. The molecular mechanism whereby *C. albicans* senses galactose and subsequent signaling that activates the expression of *GAL* genes are far less clear. Here, we find that Rep1, a DNA-binding scaffold protein in *C. albicans*, recognizes galactose via a direct physical interaction. The interaction between Rep1 and galactose allows the recruitment of a transcriptional activator Cga1 (Candida galactose gene activator, orf19.4959) to the *GAL* gene promoter, and then initiation of transcription can occur.

## Results

### Rep1 is required for galactose signaling in *C. albicans*

Rep1 was previously shown to be the transcription factor of GlcNAc signaling in *C. albicans* via the recruitment of GlcNAc sensor Ngs1 (*Su et al., 2016*). Given the intriguing observation that GlcNAc can induce the galactose metabolic genes in *C. albicans* (*Kamthan et al., 2013*), it is possible that the key regulators involved in GlcNAc signaling are also important for galactose utilization. In accord with the results from previous studies (*Su et al., 2016*; *Min et al., 2018*), deleting *NGS1* did not display a significant growth defect on galactose as the sole carbon source (*Figure 1—figure supplement 1*). In contrast, the *rep1* mutant cells grew much slower than wild type cells under the same condition (*Figure 1A*). *REP1* is not required for growth on glucose or glycerol (*Figure 1A*).To test if Rep1 regulates galactose signaling, we measured the expression of *GAL* genes. The expression of *GAL1* and *GAL10* in the presence of galactose was dramatically reduced in the *rep1* mutant strain (*Figure 1B*). The defects of *rep1* mutants could be rescued by a wild-type *REP1* under the *ADH1* promoter (*Figure 1A and B*). Although *REP1* was essential for the induction of *GAL* genes in response to galactose, disrupting *REP1* resulted in a slight increase in the expression of *GAL1* and *GAL10* in glycerol, and the increase could be erased by adding back a wild-type *REP1* into the *rep1* mutant

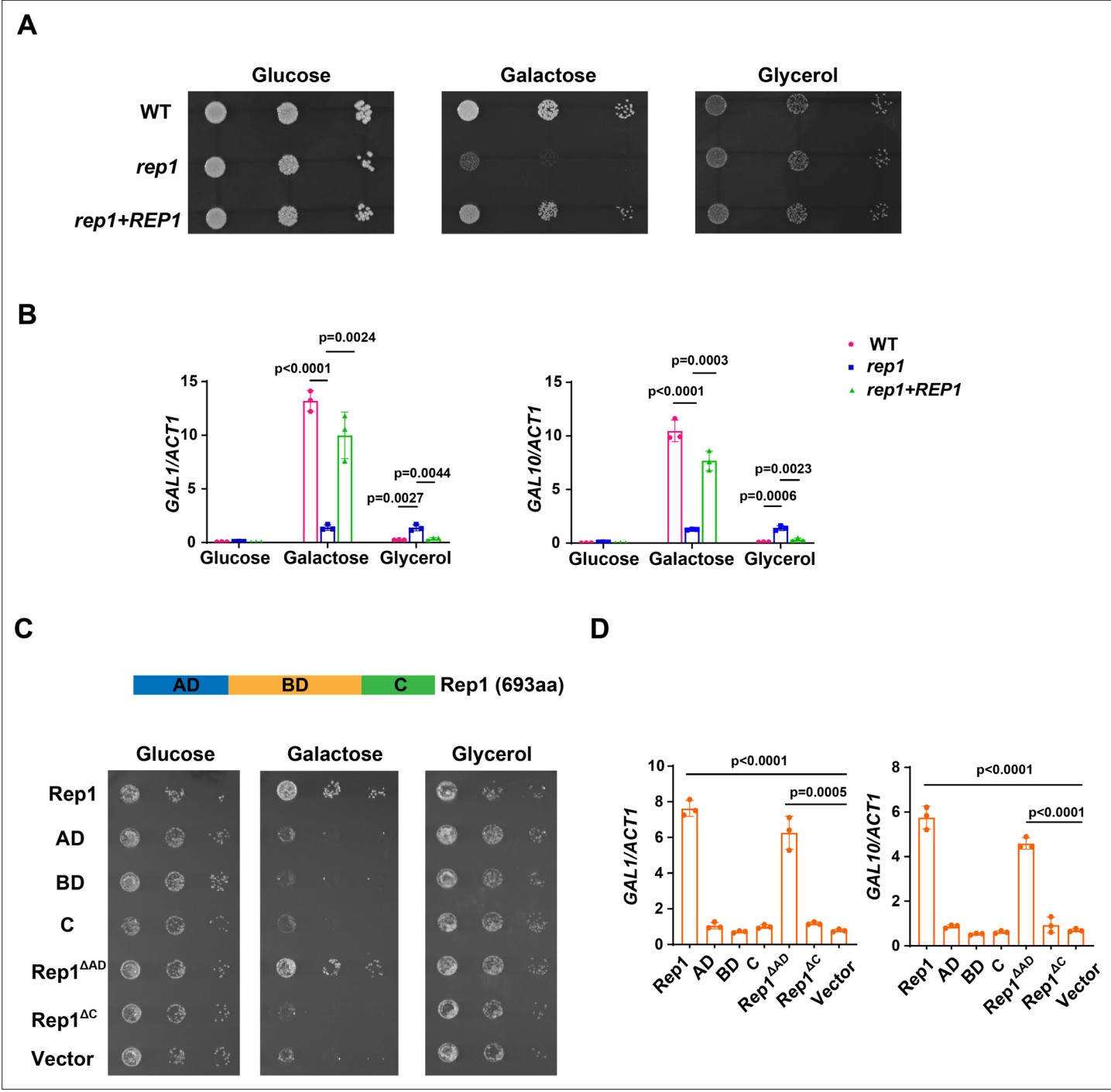

**Figure 1.** Rep1 is essential for galactose utilization in *C. albicans*. (**A**) Cells of wild type, *rep1* mutant or *rep1* mutant carrying a wild-type copy of *REP1* were diluted 10-fold and spotted onto YNB solid medium containing 25 mM of the indicated sugar. Photographs were taken after 40 hr of growth at 30 °C. (**B**) qRT-PCR analysis of galactose catabolic genes *GAL1* and *GAL10* upon galactose induction in the same strains shown in (**A**). Cells were grown in liquid SC medium with 2% galactose, glucose or glycerol for 2 hr at 30 °C for RNA extraction. (**C**) Domain structure of Rep1. The transcriptional activation domain (AD) is colored in blue, the DNA binding domain (BD) is colored in orange and the domain located in C-terminus of Rep1 is colored in green. *rep1* mutant cells carrying a wild type copy of Rep1, Rep1truncationsAD, BD, C, as well as Rep1$^{\Delta C}$, Rep1$^{\Delta AD}$ or vector alone were serially diluted 10-fold and spotted onto YNB solid medium containing 2.5 mM galactose, glucose or glycerol. Graphs were taken after 40 hr of growth at 30 °C. aa: amino acids. (**D**) The AD domain is unnecessary for Rep1-mediated induction of GAL genes. qRT-PCR analysis of *GAL1* and *GAL10* upon galactose induction in the same strains shown in (**C**). (**A and C**) Representative images of three independent experiments are shown. (**B and D**) Data shown as means ± SD of three independent experiments. Statistical analysis was performed using an unpaired two-tailed Student's *t*-test.

*Figure 1 continued on next page*

*Figure 1 continued*

The online version of this article includes the following source data and figure supplement(s) for figure 1:

**Source data 1.** Raw xlsx files used for analysis of the dataset.

**Figure supplement 1.** *ndt80* or *ngs1* mutant did not display a significant defect on galactose utilization in *C. albicans*.

**Figure supplement 2.** Rtg1 or Rtg3 is not necessary for galactose utilization and *Rep1-mediated galactose signaling* in *C. albicans*.

**Figure supplement 2—source data 1.** Raw xlsx files used for analysis of the dataset.

**Figure supplement 3.** The chimera construct Ndt80-Rep1$^C$ fails to rescue the defect of *rep1* mutant in galactose signaling.

**Figure supplement 3—source data 1.** Raw Western blot images with labeled band of interest.

(*Figure 1B*). This result resembled a previous report that Rep1 represses *MDR1* expression in *C. albicans* (*Chen et al., 2009*).

A previous study reported that the heterodimeric helix-loop-helix transcription factors Rtg1 and Rtg3 were implicated in the regulation of galactose catabolism in *C. albicans* (*Dalal et al., 2016*). Although both *rtg1* and *rtg3* mutants exhibited growth defect on galactose media supplemented with Antimycin A (*Dalal et al., 2016*), a chemical to prevent respiration, they grew normally on media supplemented with galactose as the sole carbon source without Antimycin A (*Figure 1—figure supplement 2A*). Likewise, no defect was observed in the induction of *GAL1* and *GAL10* in either *rtg1* or *rtg3* mutant in response to galactose (*Figure 1—figure supplement 2B*). The strain knocked out for *CPH1*, another transcriptional regulator previously implicated in galactose metabolism in *C. albicans* (*Martchenko et al., 2007*), did not display a galactose-specific growth defect either (*Figure 1—figure supplement 2A*). We also performed genetic screens with both a knockout library of 674 unique genes in *C. albicans* (*Noble et al., 2010*) and the GRACE library, a non-redundant library containing a total of 2357 different mutants (*Roemer et al., 2003*), where none of the mutant was found to be unable to grow on galactose, underscoring the exclusive role of Rep1 for galactose utilization in *C. albicans*. Thus, Rep1 is the most pivotal regulator identified to date for galactose signaling in *C. albicans*.

Rep1 is defined as a Ndt80 family transcription factor as it contains a DNA binding domain of this family. In addition to the conventional transcriptional activation domain (AD) and DNA binding domain (BD) of a transcription factor, a carboxyl-terminal extension (C) is emerged in Rep1 protein (*Figure 1C*), which distinguishes Rep1 from other Ndt80 transcription factors. Unlike Rep1, Ndt80 is not required for the growth on galactose (*Figure 1—figure supplement 1*). To determine whether the functional specificity of Rep1 regarding galactose utilization is the presence of C domain, we designed a chimeric construct in which the C domain from Rep1 was fused to Ndt80 protein at its C-terminus (*Figure 1—figure supplement 3A*). This chimeric construct failed to restore the growth defect of the *rep1* mutant (*Figure 1—figure supplement 3B*), suggesting that, except for the C domain, regulation of the transcriptional activity of Rep1 upon galactose induction also occurs elsewhere in the protein. The protein level of chimeric construct is comparable to that of wild-type Rep1 (*Figure 1—figure supplement 3C*). We therefore dissected the role of the individual Rep1 domains. As expected, removing the C domain from Rep1 (Rep1$^{\Delta C}$) failed to restore the growth defect of the *rep1* mutant on galactose (*Figure 1C*). The C domain alone had no function either (*Figure 1C*). To our surprise, the transcriptional activation domain of Rep1 is not necessary for its galactose responsiveness as introducing a truncation lacking this domain (Rep1$^{\Delta AD}$) into the *rep1* mutant showed a comparable growth rate on galactose to that of full-length Rep1 (*Figure 1C*). Rep1$^{\Delta AD}$ could also induce the expression of *GAL* genes in response to galactose, although a slightly lower expression level was observed in this truncation than that of its full-length counterpart (*Figure 1D*). These results indicate that the transcriptional activation domain of Rep1 is not necessary for galactose signaling in *C. albicans*.

## Rep1 binding to galactose is essential for the induction of *GAL* genes

How does Rep1 provide galactose responsiveness? Galactose was unable to increase the protein level of Rep1 (*Figure 2—figure supplement 1A*). In vivo chromatin immunoprecipitation (ChIP) showed that Rep1 bound to the divergent promoter region shared by *GAL1* and *GAL10* in a galactose-independent manner (*Figure 2A*). In addition, GFP-Rep1 expressed from the *RHO1* promoter accumulated in the nucleus regardless of the presence of galactose (*Figure 2—figure supplement 1B*). Therefore, Rep1-mediated transcriptional activation regarding galactose signaling cannot be regulated by the control

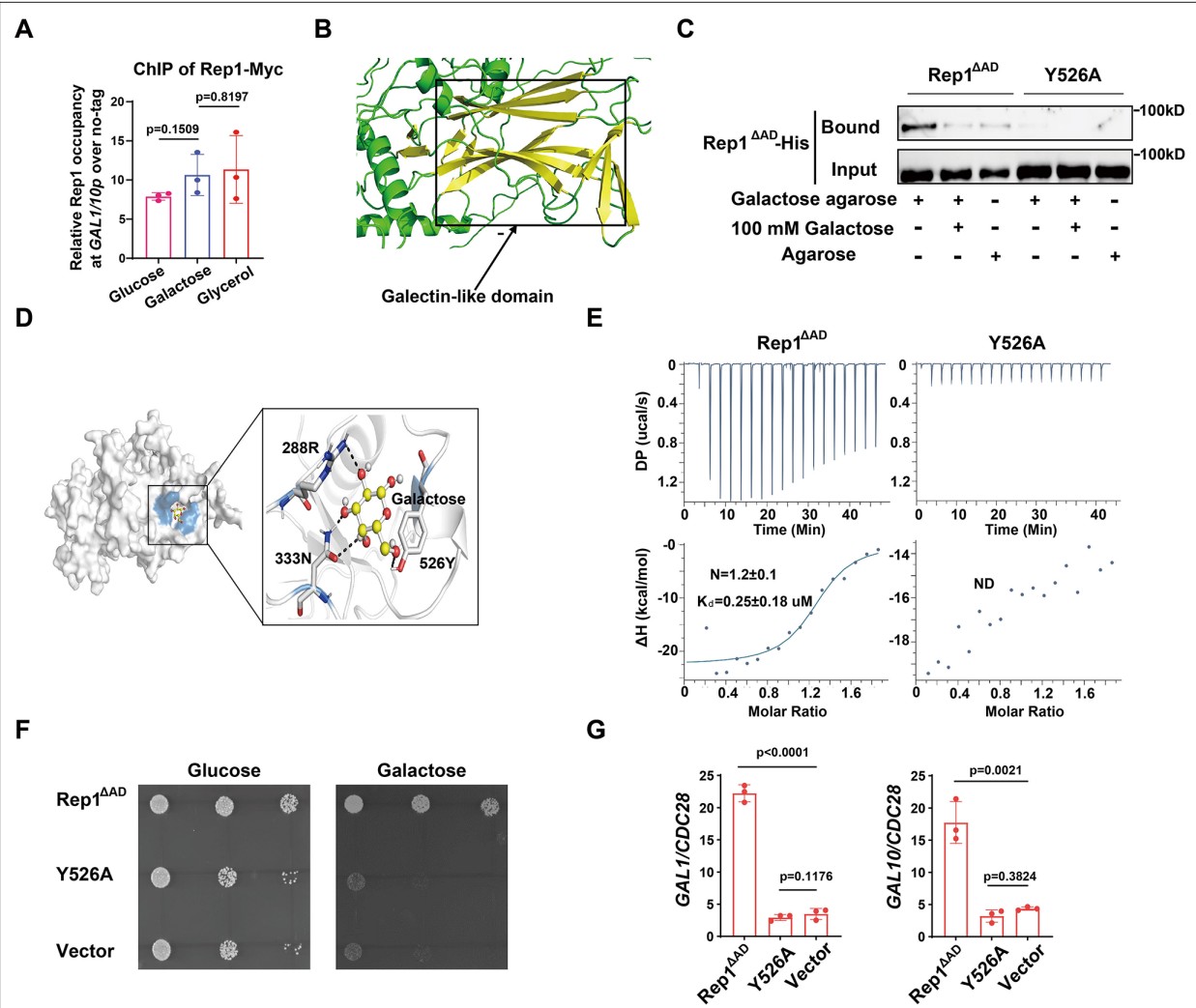

**Figure 2.** Binding of galactose to Rep1 is required for galactose signaling. (**A**) Rep1 is constitutively presented at the *GAL1-GAL10* promotor. Overnight culture of wild type cells carrying Rep1-Myc and untagged control was inoculated into SC medium containing 2% indicated sugar at 30 °C for 4 hr for the ChIP experiment. The enrichment is presented as a ratio of *GAL1-GAL10* promotor IP (bound/input) over the control region *ACT1* IP (bound/input) and is then normalized to the untagged strain. The ChIP data showed the average of three independent experiments with error bars representing the SD. (**B**) Structure model for Rep1 predicted by AlphaFold2. A galectin-like fold with the characteristic two twisted β-sheets separated by a cleft is shown in Rep1. (**C**) Rep1$^{\Delta AD}$ directly binds to galactose. Recombinant Rep1$^{\Delta AD}$ and its mutation in Y526 were incubated with Galactose-agarose or control beads at 4 °C overnight in the absence or presence of 100 mM galactose. The input and bound fractions were analysed on Western blots probed with His antibody. Representative blots of three independent experiments are shown. (**D**) Molecular docking of galactose onto Rep1$^{\Delta AD}$. The highest-ranked docked galactose is shown in the ball-and-stick model, and the protein is shown with a transparent surface. The inset shows how galactose interacts with Rep1$^{\Delta AD}$ residues. The galactose and the Rep1$^{\Delta AD}$ residues are shown in ball-and-stick and stick models, respectively. Hydrogen bond interactions are shown in dashed lines. (**E**) ITC binding curves using recombinant Rep1$^{\Delta AD}$ or its Y526A mutation and galactose (n=3; mean ± SD). (**F**) Disrupting the binding of Rep1 to galactose leads to a defect in galactose utilization in *C. albicans*. *rep1* mutant cells carrying vector alone, Rep1$^{\Delta AD}$ or its Y526 mutation were serially diluted 10-fold and spotted onto YNB solid medium containing 25 mM galactose or glucose. Graphs were taken after 40 hr of growth at 30 °C. Representative images of three independent experiments are shown. (**G**) qRT-PCR analysis of *GAL1* and *GAL10* upon galactose induction in the same strains shown in (**F**). A & G, Data shown as means ± SD of three independent experiments. Statistical analysis was performed using an unpaired two-tailed Student's *t*-test.

The online version of this article includes the following source data and figure supplement(s) for figure 2:

**Source data 1.** Raw xlsx files used for analysis of the dataset.

**Source data 2.** Raw Western blot images with labeled band of interest.

**Figure supplement 1.** The expression level, DNA-binding property or localization of Rep1 is not regulated by carbon source.

**Figure supplement 1—source data 1.** Raw Western blot images and Coomassie stained gel images with the labeled band of interest.

of its expression, DNA-binding properties or localization, but, instead, changes in the Rep1 protein per se must occur in response to galactose. Rep1 was predicted to comprise a putative galectin-like fold by AlphaFold2 (*Tunyasuvunakool et al., 2021*; *Jumper et al., 2021*; *Figure 2B*). Given that galectins were discovered based on their galactoside binding activity, we hypothesize that Rep1 possesses the ability to interact with galactose.To test this possibility, we purified recombinant Rep1$^{\Delta AD}$-His. We removed the transcriptional activation domain, which is not necessary for function (*Figure 1C*), to improve solubility. Agarose conjugated with galactose, but not control agarose, was able to pull down recombinant Rep1$^{\Delta AD}$-His (*Figure 2C*). The binding of galactose-agarose with Rep1$^{\Delta AD}$-His exhibited close to complete abolishment when free galactose was added to the cell lysate before incubation with galactose-agarose (*Figure 2C*).

To understand the molecular interactions between Rep1$^{\Delta AD}$ and galactose, we performed molecular docking using AutoDock Vina (*Trott and Olson, 2010*). The docking pose with the lowest binding free energy of –4.2 kcal/mol shows that galactose binds to a shallow pocket on Rep1$^{\Delta AD}$ (*Figure 2D*). Three Rep1$^{\Delta AD}$ residues are involved in a network of interactions with galactose. Among them, Y526 is supposed to be central to the interaction as it emerges proximal to galactose and participates in both hydrogen bonding interactions and hydrophobic interactions with galactose (*Figure 2D*). We therefore replaced Y526 with alanine in Rep1$^{\Delta AD}$ and found that this mutation did not change the protein level of Rep1$^{\Delta AD}$ both in vivo and in vitro (*Figure 2—figure supplement 1C and D* & 1D). As shown in *Figure 2C*, substitution of Y526 in Rep1$^{\Delta AD}$-His with alanine almost completely abrogated interactions with galactose. Isothermal titration calorimetry (ITC) further confirmed a direct binding between Rep1$^{\Delta AD}$-His and galactose, with the dissociation constant (Kd) measured as ~0.25 µM (*Figure 2E*), whereas an interaction between Rep1 and glucose, GlcNAc or arabinose could not be detected (*Figure 2—figure supplement 1E*), indicating that this binding is specific to galactose. Consistent with the immunoprecipitation analyses, the Y526A mutant showed no detectable interaction with galactose in an ITC assay (*Figure 2E*). Notably, disrupting the interaction between Rep1$^{\Delta AD}$ and galactose by mutating Y526 to alanine resulted in a severe growth defect on galactose (*Figure 2F*), as well as a defect in the induction of *GAL* genes expression (*Figure 2G*). Thus, Rep1 recognizes galactose via a direct physical interaction, which is crucial for galactose signaling.

## Recognition of galactose by Rep1 enables the recruitment of Cga1 for transcriptional activation

Next, we investigate how the interaction between Rep1 and galactose induces the expression of *GAL* genes. The fact that the transcriptional activation domain of Rep1 is not necessary for the induction of *GAL* genes promoted us to determine whether Rep1 possesses a transcriptional activity. To this end, we employed a one-hybrid system with *C. albicans* (*Russell and Brown, 2005*), in which *REP1* was fused to the *lexA* DNA-binding domain and the readout of transcriptional regulation was done using a *lex* operator containing *lacZ* reporter gene construct.The level of reporter gene expression was assayed by using quantitative analysis of β-galactosidase activity. As shown in *Figure 3A*, we found no activating activity of Rep1. As a control, this assay confirmed that Ndt80 acts as a transcriptional activator (*Figure 3A*).

Given that Rep1 does not itself possess a transcriptional activity, we hypothesize that Rep1-mediated transcriptional activation upon galactose binding should be mediated by its interaction partner(s). As such, we proposed a putative model for the molecular mode-of-action regarding the activation of the *GAL* genes: the interaction between Rep1 and galactose is a prerequisite for the recruitment of a transcriptional activator (X) to the *GAL* genes promotor, enabling transcriptional activation to occur (*Figure 3B*). We therefore identified proteins that interact with Rep1 specifically in the presence of galactose. GFP or GFP-Rep1 was immunoprecipitated from wild-type *C. albicans* cells in medium with galactose, glucose or glycerol as the sole carbon source, and the samples were analyzed by mass spectrometry. The identified proteins that were unique or highly enriched in the GFP-Rep1 samples in galactose from three independent repeats are shown in *Figure 3C*. Among the top three hits, we focused on Orf19.4959, which is described to possess potential DNA-binding transcription factor activity in Candida Genome Database (CGD). We then deleted this gene in *C. albicans* and found that cells lacking Orf19.4959 were unable to utilize galactose (*Figure 3D*), and failed to induce the expression of *GAL* genes as well (*Figure 3E*), which resembled the phenotype of the *rep1* mutant. The expression level of Orf19.4959 was largely unchanged in galactose compared

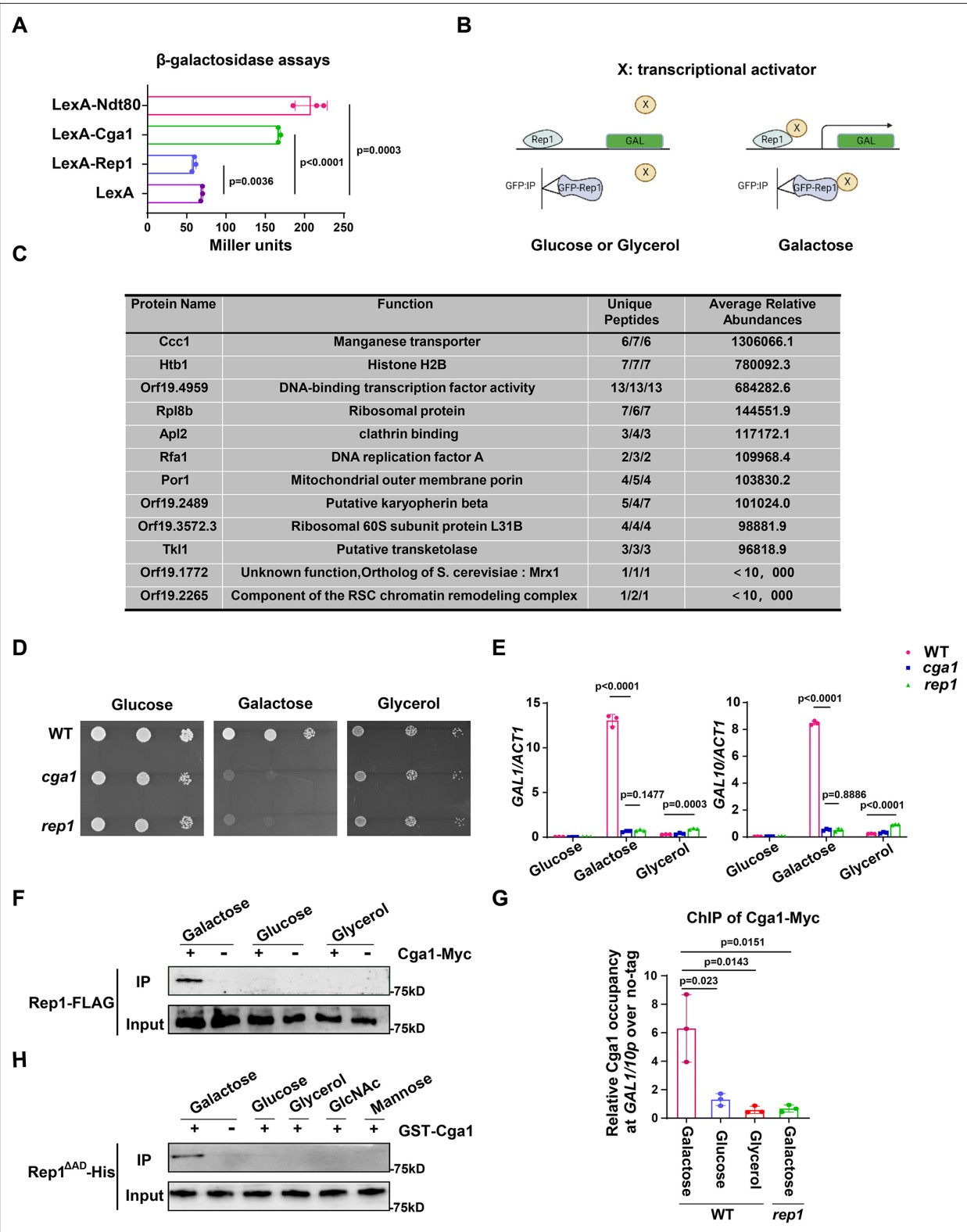

**Figure 3.** The recognition of galactose by Rep1 enables the recruitment of Cga1 to activate transcription. (**A**) Rep1 does not possess the ability to activate transcription, but Cga1 has it. Liquid β-galactosidase assays (in Miller units) of the indicated fusions. (**B**) A putative model for Rep1-mediated induction of the *GAL* genes is shown. Protein X represents a transcriptional activator. (**C**) Putative Rep1 interaction partners specifically in the presence of galactose identified by Mass Spectrometry. Data were obtained from three independent experiments, and the number of unique peptides from each experiment shown together with an average ion score. (**D**) Cells of indicated strains were diluted 10-fold and spotted onto YNB solid medium containing

*Figure 3 continued on next page*

*Figure 3 continued*

25 mM of galactose, glucose or glycerol. Photographs were taken after 40 hr of growth at 30 °C. (**E**) Cga1 is essential for galactose-inducible expression of *GAL* genes. Data shown as means ± SD of three independent experiments.(**F**) Rep1 interacts with Cga1 specifically in galactose in vivo. Overnight culture of wild type cells carrying Rep1-FLAG and Cga1-Myc or Rep1-FLAG alone was diluted 1:50 into SC medium with 2% glucose, galactose or glycerol at 30 °C for 4 hr. Protein lysates were subjected to immunoprecipitation with anti-Myc antibody, and theprecipitated proteins were probed with anti-FLAG antibody. As an inputcontrol, cell lysates were analysed by Western blotting with the anti-FLAGantibody. (**G**) Rep1 recruits Cga1 to the *GAL* gene promoter in a galactose-dependent manner. Cells of wild type or *rep1* mutant strain carrying Cga1-Myc were diluted at 1:50 into SC medium containing indicated sugars at 30 °C for 4 hr. The enrichment is presented as a ratio of *GAL1-GAL10* promotor IP (bound/input) over the control region *ACT1* IP (bound/input) and is then normalized to the untagged strain.Data shown as means ± SD of three independent experiments. (**H**) Rep1 directly interacts with Cga1 in the presence of galactose. Recombinant GST-Cga1 and Rep1$^{\Delta AD}$-His were incubated with Glutathione-agarose in the presence of glucose, galactose or glycerol at 4 °C for 2 hr. Samples were assayed by immunoblot with the anti-His antibodies. (**A, E and G**) Data shown as means ± SD of three independent experiments. Statistical analysis was performed using an unpaired two-tailed Student's *t*-test. (**F and H**) Representative blots or images of three independent experiments are shown.

The online version of this article includes the following source data and figure supplement(s) for figure 3:

**Source data 1.** Raw xlsx files used for analysis of the dataset.

**Source data 2.** Raw Western blot images with the labeled band of interest.

**Source data 3.** The source file for the table.

**Figure supplement 1.** Cga1 is constitutively expressed and accumulated in the nucleus regardless of carbon source.

**Figure supplement 1—source data 1.** Raw xlsx files used for analysis of the dataset.

**Figure supplement 1—source data 2.** Raw Coomassie stained gel images with the labeled band of interest.

to that in glucose (*Figure 3—figure supplement 1A*), and it constitutively accumulated in nucleus regardless of carbon sources (*Figure 3—figure supplement 1B*), both of which resembled to that of Rep1. Given the essential role of Orf19.4959 on galactose signaling and its transcriptional activation property (*Figure 3A*), we therefore designated it as Cga1 (Candida galactose gene activator). In addition to Cga1, we noticed that the other two hits: Orf19.1772 and Orf19.2265, which showed lower abundance in GFP-Rep1 immunoprecipitated samples in galactose (*Figure 3C*), may possess potential transcriptional regulatory activity. As shown in *Figure 3—figure supplement 1C, D*, neither of them is necessary for galactose utilization.

We next tested whether Rep1 interacts with Cga1 using an in vivo immunoprecipitation assay. Cga1-Myc, but not no tag control, immunoprecipitated endogenous Rep1-FLAG from lysate of cells grown in galactose (*Figure 3F*). No interaction was detected between Cga1-Myc and Rep1-FLAG in glucose or glycerol (*Figure 3F*). ChIP experiment showed that, unlike Rep1 that constitutively presented on *GAL* genes promoter, the association of Cga1 to the *GAL1-GAL10* promoter could be only detected from galactose-grown cells in a Rep1-dependent manner (*Figure 3G*). These results indicate that Cga1 itself is unable to bind the *GAL* promoters but could be recruited by Rep1 in a manner dependent on the presence of galactose. Consistent with the results that Rtg1 and Rtg3 are dispensable for galactose utilization in *C. albicans* (*Figure 1—figure supplement 2A, B*), *rtg1* or *rtg3* mutant strain did not display any defect on promoter binding of Rep1 and subsequent galactose-induced recruitment of Cga1 to *GAL* gene promoter (*Figure 1—figure supplement 2C*). To establish whether Rep1 is directly responsible for the recruitment of Cga1 upon galactose binding, GST-Cga1, or GST as a negative control, was expressed in *E. coli* and purified (*Figure 3—figure supplement 1E*). GST pull down using GST-Cga1, but not GST alone, readily detected interaction with recombinant Rep1$^{\Delta AD}$-His only when galactose was added, but not when glucose, glycerol, GlcNAc or mannose added (*Figure 3H*), indicating that the interaction between Rep1 and Cga1 specifically depends on the presence of galactose. These in vivo and in vitro immunoprecipitation experiments demonstrate that Rep1 directly recruits Cga1 to the *GAL* gene promoters upon galactose binding and Cga1 is responsible for subsequent transcriptional activation.

## The AD domain of Rep1 is responsible for recruitment of Ngs1 for GlcNAc-responsive transcription

It has been shown that Rep1 recruits Ngs1, the sensor-transducer of GlcNAc signaling, to target promoters (*Su et al., 2016*). Although the AD domain of Rep1 is not necessary for galactose signaling, it is essential for cellular responses to GlcNAc, as removal of this domain resulted in a growth defect

in GlcNAc (*Figure 4A*) and abolished GlcNAc-induced *GAL* gene expression (*Figure 4B*). Furthermore, we found that Ngs1 was also necessary for the *GAL* gene induction in response to GlcNAc (*Figure 4B*). These observations led us to hypothesize that the AD domain of Rep1 is responsible for the recruitment of Ngs1 to activate GlcNAc-responsive transcription. Indeed, no interaction was detected between Rep1-Myc and FLAG-Ngs1 in a co-immunoprecipitation assay following deletion of the AD domain from Rep1 (*Figure 4C*). These results indicate that Rep1 employs different domains to recruit Cga1 and Ngs1, which links galactose and GlcNAc catabolism in *C. albicans*.

## Identification of Rep1 *cis*-regulatory motif and target genes

Both Rep1 and Ndt80 belong to the Ndt80-like transcription factor family. As described above, Rep1 regulates galactose and GlcNAc catabolism (*Figure 1A and B*, *Su et al., 2016*), whereas Ndt80 is a master regulator of biofilm formation (*Nobile et al., 2012*). The difference in phenotype should be due to a difference in the target genes between these regulators,although they are thought to share a conserved DNA binding domain. To this end, we identified the genes directly regulated by Rep1 using chromatin immunoprecipitation of Myc-tagged Rep1, as well as an untagged control, in SC medium supplied with galactose as carbon sourcefollowed by high-throughput sequencing (ChIP-Seq). With a $Q$ value cutoff of 0.05, an adenine-rich motif was found in 74 out of 104 sequences by using the FIMO tool from MEME's suite (*Figure 4D*), which differs from that of Ndt80 reported previously (CACAAA) (*Nocedal et al., 2017*).

After elimination of the intragenic and downstream peaks, 108 genes showed enrichment in IP versus no tag control in the upstream region. Both galactose catabolic genes (*GAL1*, *7&10*) and GlcNAc catabolic genes (*DAC1* and *NAG1*) were identified as direct targets of Rep1 in our ChIP-Seq assay, supporting for the validity of the full genome ChIP data. We then subjected the target genes of Rep1 to pathway analysis using Gene Ontology (GO) and showed enrichment in diverse processes, including response to extracellular stimulus, carbohydrate metabolic process, organelle disassembly (*Figure 4E*). Although the ChIP-Seq experiment revealed the regions where Rep1 binds, it does not indicate whether these binding events are associated with differences in gene transcription. To assess the relationship of Rep1 binding and transcription, we performed qRT-PCR analyses under galactose or glycerol condition. Rep1-mediated transcriptional activation in response to galactose was observed inGlcNAc catabolic genes *DAC1* and *NAG1* (*Figure 4—figure supplement 1*). However, the induction ratio of these genes in galactose was much lower (~10-fold) than that in GlcNAc (>100-fold) (*Su et al., 2016*). Except for GlcNAc and galactose catabolic genes, an increase in the expression of other Rep1 target genes we tested was observed when *REP1* was deleted (*Figure 4—figure supplement 1*), in agreement with an inhibitory role of Rep1 in transcription. These findings suggest that Rep1 is bound to its target promoters irrespective of whether those genes are transcriptionally active. Therefore, it is conceivable that Rep1 provides a scaffold to recruit different transcriptional regulators, thereby enabling transcriptional activation in response to environmental or developmental signals.

## Conservation of Rep1-mediated galactose sensing mechanism in fungi

Having identified Rep1-Cga1 mode of regulation for the *GAL* regulon in *C. albicans*, we wonder whether it is exploited wildly in fungi. To this end, we performed BLAST searches to analyze the distribution of Rep1 orthologs in fungi and found that Rep1 orthologs are presented in CTG clade, and in *Aspergillus nidulans* belonging to Eurotiales clade (*Figure 4F*). Notably, the distribution pattern of Cga1 almost mirrors to that of Rep1 (*Figure 4F*), suggesting that the Rep1-Cga1 mode of regulation for galactose utilization should restrict to species most closely related to *Candida branches. To further determine whether the function of Rep1 otholgs is conserved in CTG species, we examined the ability of Rep1 in Candida tropicalis or Candida parapsilosis to complement the galactose utilization defect in C. albicansrep1 mutant. The cross-species experiment showed that ectopically expressed CtREP1 or CpREP1 rescued the growth defect of Carep1 mutant on galactose (Figure 4—figure supplement 2A*), and partially rescued the defects in the induction of galactose catabolic genes (*Figure 4—figure supplement 2B*). Similarly, introducing a WT copy of *CtCGA1* or *CpCGA1* into the *Cacga1* mutant could also rescue its defects in galactose utilization (*Figure 4—figure supplement 2C and D*). The results suggest that galactose sensing and signal transduction mediated by Rep1 and Cga1 is conversed in CTG species.

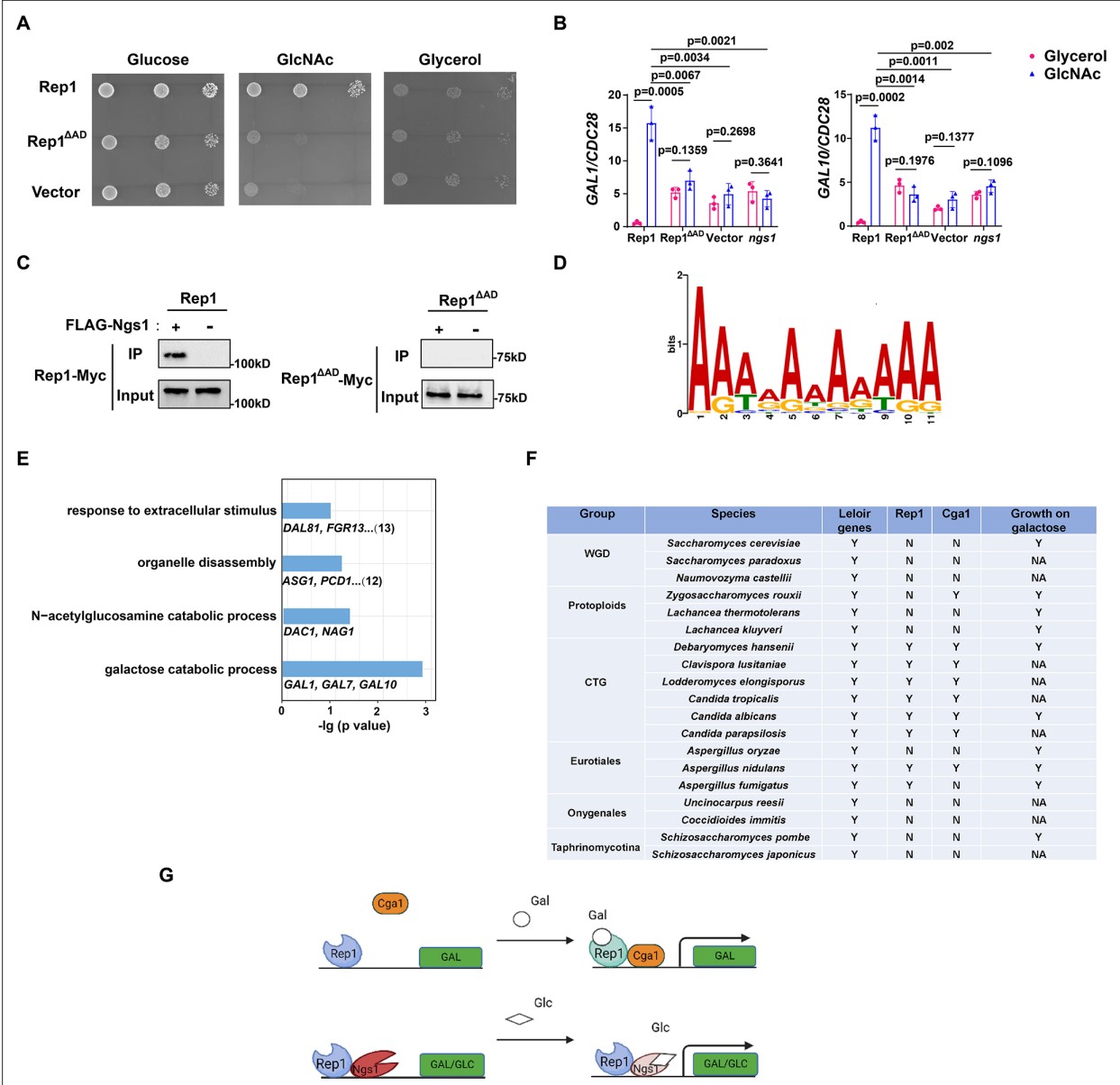

**Figure 4.** Rep1 provides a scaffold that regulates target genes via its associated partners. (**A**) The AD domain of Rep1 is essential for GlcNAc signaling. Cells of the *rep1* mutant carrying Rep1,Rep1[ΔAD] or vector alone were spotted onto YNB solid medium containing 2.5 mM glucose, glycerol or GlcNAc. Photographs were taken after 40 hr of growth at 30 °C. Representative images of three independent experiments are shown. (**B**) Both the AD domain of Rep1 and Ngs1 are required for the induction of *GAL1/GAL10* in GlcNAc. Data shown as means ± SD of three independent experiments.Statistical analysis was performed using an unpaired two-tailed Student's *t*-test.(**C**) The AD domain of Rep1 is required for the recruitment of Ngs1. Protein lysates from wild type cells carrying Rep1-Myc, FLAG-Ngs1 & Rep1-Myc, Rep1[ΔAD]-Myc, or FLAG-Ngs1 &Rep1[ΔAD]-Myc were subjected to immunoprecipitation with anti-FLAG antibody, and the precipitated proteins were probed with anti-Myc antibody. As an input control, cell lysates were analysed by Western blotting with the anti-Myc antibody. Representative blots of three independent experiments are shown. (**D**) The *cis*-regulatory motif most highly enriched at locations of Rep1 in *C. albicans*. Motifs were generated using MEME. (**E**) The major terms after performing GO enrichment analysis. Bar graphs represent the corrected p-value. Representative genes for each GO term are listed below and the number of genes is shown on the right. (**F**) Presence–absence of Leloir genes, key regulators (Rep1, Cga1) and growth on galactose media. Y, present; N, absent; NA, not available. Growth data were obtained from the review article (***Choudhury and Whiteway, 2018***). (**G**) Model of Rep1-mediated transcriptional activation in response to galactose (top) or GlcNAc (bottom). Rep1 recognizes galactose via directly physical interaction, enabling the recruitment of Cga1 to activate transcription. While Rep1 constitutively interacts with GlcNAc sensor Ngs1, the transcriptional activation upon GlcNAc induction is achieved by promoter histone acetylation via GlcNAc binding to Ngs1.

The online version of this article includes the following source data and figure supplement(s) for figure 4:

**Source data 1.** Raw xlsx files used for analysis of the dataset.

*Figure 4 continued on next page*

*Figure 4 continued*

**Source data 2.** Raw Western blot images with the labeled band of interest.

**Source data 3.** The source file for the table.

**Figure supplement 1.** Rep1 binds to its target promoters irrespective of whether those genes are transcriptionally active.

**Figure supplement 1—source data 1.** Raw xlsx files used for analysis of the dataset.

**Figure supplement 2.** Conservation of Rep1-Cga1 for galactose utilization in CTG species.

**Figure supplement 2—source data 1.** Raw xlsx files used for analysis of the dataset.

## Discussion

In the present study, we discover a DNA-binding scaffold protein Rep1 that functions as a galactose sensor in *C. albicans*. The recognition of galactose by Rep1 via a direct physical interaction brings about the recruitment of a transcriptional activator Cga1, thereby activating transcription of the *GAL* genes (*Figure 4G*). Following the Gal4 mode in Saccharomycetaceae clade, our discovery of Rep1-Cga1 mode in *C. albicans* provides a second transcriptional network that dictates the induction of *GAL* regulon in yeasts.

Rep1 is defined as a Ndt80 family transcription factor, which preserves the DNA-binding property but loses the ability to activate transcription (*Figures 2A and 3A*). It is essential for the induction of both genes that produce galactose-metabolizing enzymes and GlcNAc catabolic enzymes in *C. albicans*. Rep1-mediated transcriptional activation is achieved by either galactose-inducible recruitment of Cga1 to *GAL* promoters, or by Ngs1-induced promoter chromatin acetylation upon GlcNAc binding (*Figure 4G*; *Su et al., 2016*). Protein binding of Cga1 to Rep1 relies on the presence of galactose. However, the interaction between Ngs1 and Rep1 seems independent of carbon sources (*Su et al., 2016*). We propose that Rep1 recruits Ngs1 to *GAL* gene promoters; the presence of GlcNAc induces transcription of *GAL* genes via Ngs1 whose *N*-acetyltranferase activity is activated upon GlcNAc binding (*Figure 4G*). This idea may help to explain why GlcNAc can induce the *GAL* gene expression in *C. albicans* and different alternative carbon sources can be assimilated simultaneously by this commensal-pathogenic fungus.Although ChIP-Seq data revealed that ~100 genes were identified as Rep1 targets, Rep1 played a minor role on the regulation of their expression under non-inducing conditions (*Figure 4—figure supplement 1*). Thus, unlike Ndt80 that itself serves as a transcriptional activator (*Figure 3A*), Rep1 provides a scaffold for recruitment of various transcriptional co-regulators, which in turn activates transcription in response to environmental cues.Through these properties, Rep1-mode of regulation may facilitate crosstalk among distinct signaling pathways, which imparts metabolic flexibility of *C. albicans* to colonize diverse host niches.

Transcriptional rewiring of metabolic pathways probably generates a great diversity of biological responses and evolutionary novelties. Yeasts occupy a great diversity of ecological niches and have therefore provided an excellent opportunity to investigate gene expression circuitries chosen by organisms living in a variety of niches. There is substantial rewiring in the regulatory networks of *GAL* pathway: from Gal3-Gal80-Gal4 mode in *S. cerevisiae* to Rep1-Cga1 mode in *C. albicans*. Neither Rep1 nor Cga1 has a clear ortholog in *S. cerevisiae* (*Figure 4F*), and the binding motif of Rep1 also differs from that of Gal4 (*Figure 4D*). This rewiring therefore occurred through changes in a combination of de novo genes in the regulatory network as well as the cis-regulatory sequences of the control regions of *GAL* genes.The orthologues of both Rep1 and Cga1 seem present only in CTG species and in very limited species in Eurotiales clade (*Figure 4F*), which raises the hypothesis that there is a variety of modes of regulation of galactose metabolism in fungi. Since the *GAL* pathway in budding yeasts has often served as a model for studying gene regulation and evolution in eukaryotes (*Johnston, 1987*; *Harrison et al., 2022*; *Hittinger et al., 2010*; *Slot and Rokas, 2010*; *Hittinger et al., 2004*), our discovery of Rep1-Cga1 mode adds significantly to the literature of transcription regulatory network for galactose utilization in fungi.

In *S. cerevisiae*, activation of the *GAL* pathway relies on Gal80, which represses Gal4 activity until Gal3 bound to galactose sequesters it in the cytoplasm. In *C. albicans*, Rep1 and Cga1 create a relatively simple signal-transduction cascade, in which galactose binding to Rep1 recruits Cga1 to initiate transcription of *GAL* regulon. What distinguishes this system from that of *S. cerevisiae* regarding network architecture is the absence of a factor (like Gal80) that represses Cga1 activity. Given that the induction of *GAL* genes occurs faster in *C. albicans* than that in *S. cerevisiae* (*Dalal*

*et al., 2016*; *Escalante-Chong et al., 2015*), our study provides evidence consistent with the idea that short cascades orchestrate rapid gene expression responses to external stimuli (*Rosenfeld and Alon, 2003*). In addition, Gal4 abundance is controlled by glucose concentration, which determines the robustness of Leloir genes induction in *S. cerevisiae*, whereas both *REP1* and *CGA1* seem constitutively expressed regardless of carbon sources (*Figure 2—figure supplement 1A* and *Figure 3—figure supplement 1A*). This discrepancy with respect to glucose repression on major regulator(s) of the *GAL* regulon likely underlies the differences in quantitative responses to *GAL* gene induction. That is, higher basal level but lower induction ratio and more graded induction in *C. albicans* (*Dalal et al., 2016*), rather than the bimodal, on-and-off mode in *S. cerevisiae* (*Ricci-Tam et al., 2021*; *Wang et al., 2015*; *Venturelli et al., 2015*).

As a commensal-pathogenic fungus, *C. albicans* must adapt to dynamic and contrasting host microenvironments such as the gastrointestinal tract, mucosal surfaces, bloodstream, and internal organs (*Brown et al., 2014*; *Alves et al., 2020*). The ability to respond sensitively and rapidly to changing nutritional conditionscould conceivably be of selective advantage (*Pradhan et al., 2019*; *Pradhan et al., 2018*; *Ballou et al., 2016*; *Lu et al., 2013*). Specific features of Rep1-Cga1 mode of action likely impart properties that may be adaptive for *C. albicans*. Our work provides insights into how *C. albicans* evolves transcriptional programs to adapt to live as commensals or to colonize and grow efficiently within its host tissues during infection where microbial competitors and host factors can cause dynamic changes in the spectrum of carbon compounds available.

# Materials and methods

## Key resources table

| Reagent type (species) or resource | Designation | Source or reference | Identifiers | Additional information |
|---|---|---|---|---|
| Gene (*Candida albicans*) | *REP1* | Candida Genome Database | CR_00,140 W | |
| Gene (*Candida tropicalis*) | *REP1* | Candida Genome Database | CTRG1_01069 | |
| Gene (*Candida parapsilosis*) | *REP1* | Candida Genome Database | CPAR2_800080 | |
| Gene (*Candida albicans*) | *CGA1* | Candida Genome Database | C1_13,320 C | |
| Gene (*Candida tropicalis*) | *CGA1* | Candida Genome Database | CTRG1_03606 | |
| Gene (*Candida parapsilosis*) | *CGA1* | Candida Genome Database | CPAR2_202690 | |
| Strain, strain background (*Escherichia coli*) | DH5α | Tsingke | TSC-C01 | |
| Strain, strain background (*Escherichia coli*) | BL21 | Tsingke | TSC-C06 | |
| Antibody | Anti-His (mouse monoclonal) | MBL | D291-3 | 1:5000 |
| Antibody | Anti-c-Myc (rabbit polyclonal) | Sigma-Aldrich | C3956 | 1:5000 |
| Antibody | Anti-FLAG (rabbit polyclonal) | Sigma-Aldrich | F7425 | 1:3000 |
| Antibody | Anti-tubulin (rat monoclonal) | BioRad | MCA78G | 1:3000 |

*Continued on next page*

*Continued*

| Reagent type (species) or resource | Designation | Source or reference | Identifiers | Additional information |
|---|---|---|---|---|
| Antibody | Anti-mouse IgG, (HRP-linked, mouse polyclonal) | BioRad | 1706516 | 1:3000 |
| Antibody | Anti-rat IgG (HRP-linked, rat polyclonal) | CST | 7077 | 1:5000 |
| Antibody | Anti-rabbit IgG, (HRP-linked, rabbit polyclonal) | Sigma-Aldrich | A3687 | 1:5000 |
| Sequence-based reagent | pBA1-F | This paper | PCR primers | CAACAACAAATACAAAAACAAAGATCT |
| Sequence-based reagent | pBA1-R | This paper | PCR primers | ATACGACTCACATAGGGCGAATTGGGTACC |
| Sequence-based reagent | pET28a-F | This paper | PCR primers | CTCACAGAGAACAGATTGGTGGATCC |
| Sequence-based reagent | pET28a-R | This paper | PCR primers | TCAGTGGTGGTGGTGGTGGTGCTCGA |
| Peptide, recombinant protein | Rep1$^{\Delta AD}$ | This paper | | purified from *E. coli* BL21 |
| Peptide, recombinant protein | Rep1$^{\Delta AD}$ Y526A | This paper | | purified from *E. coli* BL21 |
| Peptide, recombinant protein | GST-Cga1 | This paper | | purified from *E. coli* BL21 |
| Commercial assay or kit | RNeasy Minikit | Qiagen | 74204 | |
| Commercial assay or kit | GFP-Trap | Chromotek | Gta-20 | |
| Chemical compound, drug | D-Galactose | Sigma-Aldrich | G5388 | |
| Chemical compound, drug | N-Acetyl-D-glucosamine | MP | 1727589 | |
| Chemical compound, drug | D-Glucose | Sigma-Aldrich | G7021 | |
| Chemical compound, drug | Glycerol | Sigma-Aldrich | G5516 | |
| Chemical compound, drug | D-Mannose | Sigma-Aldrich | M2069 | |
| Chemical compound, drug | D-Arabinose | Aladdin | 10323-20-3 | |
| Software, algorithm | MicroCal PEAQ-ITC | Malvern Panalytical | MicroCal ORIGIN | |
| Software, algorithm | AutoDock Vina | AutoDock Vina | | |
| Software, algorithm | Grahpad prism | Grahpad prism | 8.0.3 | |
| Other | DAPI | Solarbio | C0060 | 1 mg/ml |

## Media and growth conditions

*C. albicans* strains were grown at 30 °C in YPD (2% Bacto peptone, 2% glucose, 1% yeast extract). Transformants were selected on SC medium (0.17% Difco yeast nitrogen base w/o ammonium sulfate, 0.5% ammonium sulfate and auxotrophic supplements) with 2% glucose. The ability of cells to grow was tested by spotting dilutions of cells onto YNB (0.17% Difco yeast nitrogen base w/o ammonium sulfate, 0.5% ammonium sulfate) solid media with different sugars followed by incubation at 30 °C for 40 hr. To determine the expression of galactose catabolic genes, cells were grown overnight in liquid YPD at 30 °C, pelleted, washed three times in PBS, and diluted 1:50 in SC medium with different sugars at 30 °C.

## Plasmid and strain construction

The *C. albicans* strains used in this study are listed in *Supplementary file 1A*. The plasmids used in this study are listed in *Supplementary file 1B* . Primer sequences are listed in *Supplementary file*

*1C*. SC5314 genomic DNA was used as the template for all PCR amplifications of *C. albicans* genes. DNA fragments were subcloned into vectors by Gibson assembly. *cga1* mutant was constructed by sequential gene disruption (*Wilson et al., 1999*).

## Quantitative RT-PCR

Total RNA was purified using the RNeasy Minikit and DNase-treated at room temperature for 15 min using the RNase-free DNase set (Qiagen). cDNA was synthesized using the Maxima H Minus cDNA synthesis master mix (Thermo Scientific), and qRT-PCR was performed using the iQ SYBR green supermix (Bio-Rad) in 96-well plates.

## Western blotting

Cells were grown overnight in YPD at 30 °C, pelleted, washed three times with PBS, diluted 1:50 in SC medium with different sugars, incubated for 4 hr at 30 °C, harvested by centrifugation.Cells were resuspended in 350 µl lysis buffer (50 mM Tris-HCl pH 7.5, 100 mM NaCl, and 0.1% NP-40), supplemented with 1 mM phenylmethylsulfonyl fluoride (PMSF) and protease inhibitor cocktail tablet (Roche).Cells were lysed with acid-washed glass beads (Sigma-Aldrich) using FastPrep (MP Bio, USA) at a setting of 5.0 with 4 pulses of 3-min intervals. Finally,the supernatant was obtained and quantitative proteins were separated in an 8% SDS-PAGE gel. Blots were hybridized with the following primary antibodies at room temperature for 1 hr: anti-His (D291-3, MBL, 1: 5000), anti-Myc (C3956, Sigma-Aldrich, 1:5000), anti-FLAG (T8793, Sigma-Aldrich, 1:3000), and anti-tubulin (MCA78G, Bio-Rad, 1:300). Goat anti-mouse IgG (H+L)-HRP conjugate (1706516, Bio-Rad, 1:3000) and goat anti-rat IgG (H+L)-HRP conjugate (7077, CST, 1:5000) were used as secondary antibodies.

## Chromatin immunoprecipitation

ChIP experimental procedures were performed as previously described (*Lu et al., 2011*) with modifications. Overnight cultures of *C. albicans* were diluted 1:50 into SC medium with 2% glucose/galactose/glycerol and incubated at 30 °C for 4 hr. For the IP,5 ul of anti-Myc(Sigma) antibody was used for ~2 mg of chromatin proteins in an immunoprecipitation volume of 300 µl. ChIP signals were quantified by qPCR.

## β-galactosidase assays

Protein extraction was performed as described above. Protein concentration was determined by Nanodrop 2000. The reaction was performed at 30 °C. A total of 50 µl of protein extracts was added into 950 µl Z-buffer (12 mM $Na_2HPO_4$, 8 mM $NaH_2PO_4$, 2 mM KCl, 0.2 mM $MgSO_4$), 200 µl ONPG (4 mg/ml), 2 µl β-mercaptoethanol and incubated for 60 min. Reactions were stopped by addition of 400 µl 1 M $Na_2CO_3$, then measured $OD_{420}$ of the reaction solvent. Miller Units were calculated as described previously (*Russell and Brown, 2005*). Three biological replicates were performed and error bars representing the SD.

## Protein expression and purification

*REP1^ΔAD* and the full-length *CGA1* were synthesized by GENEWIZ (Nanjing, China). Recombinant Rep1^ΔAD-His, as well as its variants, and GST-Cga1 was expressed in *E. coli* BL21.Cells were induced with 0.5 mM isopropyl-β-d-thiogalactopyranoside (IPTG) at 16 °C overnight. Pellets were resuspended in 25 mM Tris-HCl pH 8.0, 500 mM NaCl and lysed by sonication. For purification of His-tagged proteins, lysates were applied to a Ni-NTA column (Qiagen),washed with 20 column bed volumes (CV) of resuspension buffer with 50 mM imidazole, and eluted with 5 CV of elution buffer (25 mM Tris-HCl pH 8.0, 300 mM NaCl and 250 mM imidazole). For GST pull down experiments, lysates were applied to Glutathione-Agarose (Sigma),washed with 20 CV of resuspension buffer with 1% Tween-20, and eluted with 5 CV of elution buffer (10 mM reduced glutathione [Sigma], 50 mM Tris-HCl, pH 9.0). Protein purity was determined via SDS–PAGE (Fig. S4D & S5C).

## Galactose-agarose affinity chromatography

Recombinant Rep1$^{\Delta AD}$-His and its variants were incubated with 50 µl D–Galactose agarose beads (Thermo Scientific) overnight at 4 °C. The resulting beads and control beads were washed three times with TBS buffer. The bound protein was eluted from the matrix with 50 µl of 4% SDS for 30 min at room temperature. The input and bound fractions were analysed on Western blot.

## Isothermal titration calorimetry (ITC)

ITC experiments were performed at 25 °C using a MicroCal iTC200 (Malvern) in 50 mM Tris-HCl pH 7.5 plus 100 mM NaCl. Proteins were loaded into the sample cell (300 µl), and then placed in the injection syringe (70 µl) with stirring at 750 rpm. The first injection (0.4 µl) was followed by 19 injections of 2 µl. The data were fitted by a single-binding-site model using MicroCal ORIGIN software supplied with the instrument.

## Immunoprecipitation of GFP-Rep1 and mass spectrometry

Proteins were extracted as described above. GFP-TRAP (ChromoTek) was equilibrated in cold lysis buffer and incubated with the lysate for ~2 hr at 4 °C with rotation. Beads were pelleted and washed three times with 1 ml lysis buffer and bound proteins were eluted by the addition of SDS loading buffer. A total of 20 µg of the protein samples was loaded to 10% SDS-PAGE gel electrophoresis. For the identification of proteins by mass spectrometry, gels with bound protein from triplicate experiments were sent for analysis at Novogene (Beijing, China). The resulting spectra from each fraction were searched separately against protein sequence database (Uniprot) by the search engines: Proteome Discoverer 2.2 (PD 2.2, Thermo). The search parameters are set as follows: mass tolerance for precursor ion was 10 ppm and mass tolerance for product ion was 0.02 Da. To improve the quality of data analysis, the software PD 2.2 further filtered the retrieval results: Peptide Spectrum Matches (PSMs) with a credibility of more than 99% was identified PSMs. The identified protein contains at least 1 unique peptide. The identified PSMs and proteins were retained and performed with FDR no more than 1.0%.

## In vivo Co-immunoprecipitation

A total of 400 µl of cell lysate was incubated overnight at 4 °C with 2 µg of anti-Myc or anti-Flag antibody (Sigma). Protein A-Sepharose (GE) was washed three times with lysis buffer (50 mM Tris-HCl pH 7.5, 100 mM NaCl, 0.1% NP-40) and centrifuged for 30 s at 1000 rpm. An ~40 µl suspension of protein A-Sepharose was subjected to precipitation of the immunocomplex. After incubation at 4 °C for 2 hr, beads were washed three times with lysis buffer and proteins were eluted from the beads by boiling for 10 min, and were then subjected to Western blotting.

## In vitro co-immunoprecipitation

Recombinant proteins were centrifuged with ultrafiltration device (Amicon Ultra 10 K device –10,000 NMWL), 10,000xg for 20 min and were dissolved in 150 µl lysis buffer (PBS, 1% Triton X-100, pH 7.5). Protein lysates were incubated with Glutathione-Agarose (Sigma) at 4 °C for 2 hr. Beads were washed three times by 20 column CV of lysis buffer. Proteins were eluted from beads by boiling for 10 min; eluted proteins were then subjected to western blotting.

## ChIP-Seq

Protein lysates were prepared as described in ChIP experiments. Chromatin immunoprecipitation assays and ChIP-Seq data analysis were performed by IGENEBOOK Biotechnology. Immunoprecipitated DNA was used to construct sequencing libraries following the protocol provided by the NEXT-FLEX ChIP-Seq Library Prep Kit for Illumina Sequencing (NOVA-5143–02, Bio Scientific) and sequenced on Illumina Xten with PE 150 method. Trimmomatic (version 0.38) was used to filter out low-quality reads. Clean reads were mapped to the *C. albicans* genome by Bwa (version 0.7.15). Samtools (version 1.3.1) was used to remove potential PCR duplicates. MACS2 software (version 2.1.1.20160309) was used to call peaks by default parameters (bandwidth, 300 bp; model fold, 5, 50; q value, 0.05). If the summit of a peak located closest to the TSS of one gene, the peak will be assigned to that gene. MEME (http://meme-suite.org/index.html) was used to predict motif occurrence within peaks with default settings for a maximum motif length of 12 base pairs.

## Microscopy

The fluorescence signal in the cells was observed under a Leica DM2500 microscope. DAPI (4'6-diamidino-2-phenylindole) staining was applied for the visualization of the cell nucleus. Representative image was shown in each figure.

## Statistical and reproducibility

All experiments were performed with at least three biological repeats except indicated in the figure legends, and no statistical method was used to predetermine sample sizes. No data were excluded from analyses. Analyses were conducted using Graphpad Prism 8.3.0. Results are expressed as the mean ± standard deviation (SD) as indicated, and analysed using unpaired Student's $t$-test. p Values of less than 0.05 were considered statistically significant. All experiments were performed with at least three biological repeats except indicated in the figure legends, and all attempts at replication were successful. Sample allocation was random in all experiments. No blinding was performed because none of the analyses reported involved procedures that could be influenced by investigator bias.

## Acknowledgements

We thank Dr. HP Liu (University of California, Irvine) for constructive suggestions, Dr. SC Zhang (University of North Carolina at Chapel Hill) for helpful discussions, and Drs. AJP Brown (University of Exeter), SM Noble (University of California, San Francisco) and Fungal Genetics Stock Center for kindly providing Candida strains and plasmids. This work was supported by grants from National Natural Science Foundation of China (32070074 to YL, 32170089 and 81973370 to CS) and a grant from Natural Science Foundation of Hubei Province of China (2022CFB103 to CS).

## Additional information

### Funding

| Funder | Grant reference number | Author |
|--------|------------------------|--------|
| National Natural Science Foundation of China | 32070074 | Yang Lu |
| National Natural Science Foundation of China | 32170089 | Chang Su |
| National Natural Science Foundation of China | 81973370 | Chang Su |
| Natural Science Foundation of Hubei Province | 2022CFB103 | Chang Su |

The funders had no role in study design, data collection and interpretation, or the decision to submit the work for publication.

### Author contributions

Xun Sun, Conceptualization, Resources, Data curation, Software, Formal analysis, Validation, Investigation, Visualization, Methodology, Writing - original draft, Project administration, Writing - review and editing; Jing Yu, Resources, Data curation, Software, Visualization, Methodology; Cheng Zhu, Resources, Data curation, Software, Investigation, Methodology; Xinreng Mo, Dandan Yang, Data curation, Methodology; Qiangqiang Sun, Data curation, Formal analysis, Methodology; Chang Su, Conceptualization, Resources, Data curation, Formal analysis, Supervision, Funding acquisition, Validation, Investigation, Visualization, Writing - original draft, Project administration, Writing - review and editing; Yang Lu, Conceptualization, Resources, Data curation, Software, Formal analysis, Supervision, Funding acquisition, Validation, Investigation, Visualization, Methodology, Writing - original draft, Project administration, Writing - review and editing

## Author ORCIDs
Cheng Zhu (ID) http://orcid.org/0000-0003-0260-6287
Yang Lu (ID) http://orcid.org/0000-0002-3784-7577

### Decision letter and Author response
Decision letter https://doi.org/10.7554/eLife.84155.sa1
Author response https://doi.org/10.7554/eLife.84155.sa2

## Additional files

### Supplementary files
• Supplementary file 1. (A) Candida albicans strains used in this study. (B) Plasmids used in this study. (C) Primers used in this study.

• MDAR checklist

### Data availability
The mass spectrometry proteomics data are deposited to the ProteomeXchange Consortium with the dataset identifier PXD037522. The ChIP-Seq data are deposited to Dryad (https://doi.org/10.5061/dryad.tqjq2bw35). Source Data files have been provided in Figure 1-Source data, Figure 1-figure supplement 2-Source data, Figure 1-figure supplement 3-Source data, Figure 2-Source Data 1&2, Figure 2-figure supplement 1-Source data, Figure 3-Source Data 1,2&3, Figure 3-figure supplement 1-Source data 1&2, Figure 4-Source Data 1,2&3, Figure 4-figure supplement 1-Source data and Figure 4-figure supplement 2-Source data.

The following datasets were generated:

| Author(s) | Year | Dataset title | Dataset URL | Database and Identifier |
|---|---|---|---|---|
| Xun S, Jing X, Cheng Z, Xinreng M, Qiangqiang S, Dandan Y, Chang S, Yang L | 2022 | Candida albicans NDT80 family protein Rep1 mass spectrometry | http://proteomecentral.proteomexchange.org/cgi/GetDataset?ID=PXD037522 | ProteomeXchange, PXD037522 |
| Rodriguez DL | 2023 | Candida albicans transcription factor Rep1 ChIP-Seq | https://dx.doi.org/10.5061/dryad.tqjq2bw35 | Dryad Digital Repository, 10.5061/dryad.tqjq2bw35 |

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
