## [Editor Report]

This important manuscript investigates the circuitry connecting the galactose utilization regulon of the human pathogen and model organism *Candida albicans* to the sensing of galactose. In the non-pathogenic model yeast *Saccharomyces cerevisiae* this circuit represents a textbook model that rivals the lac operon as a teaching tool. Using a broad array of mainly classical approaches, this study convincingly demonstrates the transcriptional activators that are required for galactose responsive metabolic genes in *C. albicans*. The recognition of just how different the regulation of the galactose pathway across fungal species represents an important advance in our understanding of the evolution of the regulatory control of these circuits, and makes a nice addition to the textbook version of eukaryotic gene regulation.

---

## [Decision Letter]

**Decision letter after peer review:**

Thank you for submitting your article "Recognition of Galactose by a Scaffold Protein Recruits a Transcriptional Activator for the GAL Regulon Induction in *Candida albicans*" for consideration by *eLife*. Your article has been reviewed by 2 peer reviewers, and the evaluation has been overseen by a Reviewing Editor and Kevin Struhl as the Senior Editor. The following individual involved in the review of your submission has agreed to reveal their identity: Malcolm Whiteway (Reviewer #1).

Essential revisions:

1) Please provide the controls investigating the binding of other sugars to Rep1. Does glucose bind, GlucNac, and arabinose? Defining the specificity of binding would be a useful addition to the paper.

2) Please provide quantitative data on the Mass Spec hits that justify the follow-up of Orf19.4959. Were any of the other hits tested for effects on galactose growth? Apl1, Orf19.2265 looks interesting for gene control, how about Orf19.1772? Because identifying this hit is central to the story, this section should be moved from the supplemental data to the main text, and quantification provided.

3) Please improve the labeling of the figures (see Reviewer #2's constructive suggestions on this).

4) (Optional) Testing the Rep1/Cga1 requirement for galactose metabolism in some of these other species would be required to establish the generality of the model.

*Reviewer #1 (Recommendations for the authors):*

The Rep1 protein was found to be constitutively bound to the promoters of GAL1 and GAL10, and not really influenced in this binding by carbon source – a useful experiment would be to test the deletion of the different domains in this binding experiment.

Controls investigating the binding of other sugars to Rep1 are missing; does glucose bind, GlucNac, and arabinose? Defining the specificity of binding would be a useful addition to the paper.

We are not given quantitative data on the Mass Spec hits, so Orf19.4959 follow-up seems quite arbitrary; were any of the other hits tested for effects on galactose growth? Apl1, Orf19.2265 looks interesting for gene control, how about Orf19.1772? Because identifying this hit is central to the story, this section should be moved from the supplemental data, and quantification provided.

Testing the Rep1/Cga1 requirement for galactose metabolism in some of these other species would be required to establish generality of the model.

*Reviewer #2 (Recommendations for the authors):*

This was a solid paper. My one comment would be on labeling in the figures. I think glucose should be abbreviated as glucose (so people don't think of glutamate). In the figure itself use Glycol instead of Gly (so people don't think glycine). For example, in figure 4A there is plenty of room to have Gluc and GlcNAc and it would be much less confusing for the reader. Is 4B really glycerol or should it be glucose? If it is glycerol then do you have data for glycerol versus GlcNAC for 4A? Figure 4E is a bit confusing with the bubble size. Can you just list the gene number instead?

---

## [Author Response]

Essential revisions:1) Please provide the controls investigating the binding of other sugars to Rep1. Does glucose bind, GlucNac, and arabinose? Defining the specificity of binding would be a useful addition to the paper.

Thanks for the suggestion. We performed additional ITC experiments to determine whether Rep1 can bind glucose, GlcNAc or arabinose and showed that the interaction between Rep1 and glucose, GlcNAc or arabinose, indicating that this binding is specific to galactose. This result is now provided as Figure 2—figure supplement 1E.

2) Please provide quantitative data on the Mass Spec hits that justify the follow-up of Orf19.4959. Were any of the other hits tested for effects on galactose growth? Apl1, Orf19.2265 looks interesting for gene control, how about Orf19.1772? Because identifying this hit is central to the story, this section should be moved from the supplemental data to the main text, and quantification provided.

Thanks. The quantitative data on the Mass Spec hits is now included as new Figure 3C, which shows that Orf19.4959 is among the top three hits in this assay. We also deleted Orf19.1772 and Orf19.2265, which are thought to possess potential transcriptional regulatory activity, in *C. albicans*. Both of them are not necessary for galactose utilization. This result is now provided as Figure 3—figure supplement 1C and 1D.

3) Please improve the labeling of the figures (see Reviewer #2's constructive suggestions on this).

Thanks for the suggestion. We apologize for the unclear labeling. We replace all ‘Glu’, ‘Gly’ and ‘Glc’ to ‘glucose’, ‘glycerol’ and ‘GlcNAc’ in figures respectively. In Figure 4E, we list gene numbers instead of bubbles. In addition, we have added back glycerol control to Figure 4A.

4) (Optional) Testing the Rep1/Cga1 requirement for galactose metabolism in some of these other species would be required to establish the generality of the model.

In the revision, we examined the ability of Rep1/Cga1 in *Candida tropicalis* or *Candida parapsilosis*, two CTG species, to complement the GlcNAc utilization defect in *C. albicans rep1/cga1* mutant. The othologs of Rep1/Cga1 in both *C. tropicalis* and *C. parapsilosis* rescued the growth defect in the *C. albicans rep1/cga1* mutant on galactose, as well as the defects in the induction of galactose catabolic genes. These results are now provided as Figure 4—figure supplement 2 A-D, which supports our conclusion that Rep1-Cga1 regulatory mode for galactose utilization is conserved across CTG species.

Reviewer #1 (Recommendations for the authors):The Rep1 protein was found to be constitutively bound to the promoters of GAL1 and GAL10, and not really influenced in this binding by carbon source – a useful experiment would be to test the deletion of the different domains in this binding experiment.

Rep1 is characterized by a DNA-binding domain (BD) of Ndt80 transcription factor family. Moreover, both AD and C domains of Rep1 are predicted to be intrinsically disordered, whereas BD domain is well-folded. Thus, we suggest that Rep1 binding to the promoter of *GAL1* and *GAL10* relies on its BD domain.

Controls investigating the binding of other sugars to Rep1 are missing; does glucose bind, GlucNac, and arabinose? Defining the specificity of binding would be a useful addition to the paper.

See responses to essential revision point 1.

We are not given quantitative data on the Mass Spec hits, so Orf19.4959 follow-up seems quite arbitrary; were any of the other hits tested for effects on galactose growth? Apl1, Orf19.2265 looks interesting for gene control, how about Orf19.1772? Because identifying this hit is central to the story, this section should be moved from the supplemental data, and quantification provided.

See responses to essential revision point 2.

Testing the Rep1/Cga1 requirement for galactose metabolism in some of these other species would be required to establish generality of the model.

See responses to essential revision point 4.

Reviewer #2 (Recommendations for the authors):This was a solid paper. My one comment would be on labeling in the figures. I think glucose should be abbreviated as glucose (so people don't think of glutamate). In the figure itself use Glycol instead of Gly (so people don't think glycine). For example, in figure 4A there is plenty of room to have Gluc and GlcNAc and it would be much less confusing for the reader. Is 4B really glycerol or should it be glucose? If it is glycerol then do you have data for glycerol versus GlcNAC for 4A? Figure 4E is a bit confusing with the bubble size. Can you just list the gene number instead?

See responses to essential revision point 3.